# Role of Pyroptosis in Acetaminophen-Induced Hepatotoxicity

**Hartmut Jaeschke ***, **David S. Umbaugh** and **Anup Ramachandran**

Department of Pharmacology, Toxicology and Therapeutics, University of Kansas Medical Center,
Kansas City, KS 66160, USA

\* Correspondence: hjaeschke@kumc.edu; Tel.: +1-913-588-7969

**Abstract:** Acetaminophen (APAP) is a widely used pain reliever that can cause liver injury or liver failure in response to an overdose. Understanding the mechanisms of APAP-induced cell death is critical for identifying new therapeutic targets. In this respect it was hypothesized that hepatocytes die by oncotic necrosis, apoptosis, necroptosis, ferroptosis and more recently pyroptosis. The latter cell death is characterized by caspase-dependent gasdermin cleavage into a C-terminal and an N-terminal fragment, which forms pores in the plasma membrane. The gasdermin pores can release potassium, interleukin-1β (IL-1β), IL-18, and other small molecules in a sublytic phase, which can be the main function of the pores in certain cell types such as inflammatory cells. Alternatively, the process can progress to full lysis of the cell (pyroptosis) with extensive cell contents release. This review discusses the experimental evidence for the involvement of pyroptosis in APAP hepatotoxicity as well as the arguments against pyroptosis as a relevant mechanism of APAP-induced cell death in hepatocytes. Based on the critical evaluation of the currently available literature and understanding of the pathophysiology, it can be concluded that pyroptotic cell death is unlikely to be a relevant contributor to APAP-induced liver injury.

**Keywords:** acetaminophen; hepatotoxicity; pyroptosis; gasdermin; inflammasome; caspases

## 1. Introduction

Acetaminophen (N-acetyl-p-aminophenol, APAP, paracetamol) is one of the most widely used pain reliever worldwide. At therapeutic doses, the drug is considered safe, but an overdose can induce liver injury and even acute liver failure [1,2]. Because of the wide-spread availability of APAP, intentional and accidental overdosing is relatively common, especially in western countries. Thus, APAP overdose is responsible for almost 50% and 70% of all acute liver failure cases in the US and the UK, respectively [3,4]. Because of the clinical significance, understanding the mechanisms of APAP-induced liver injury has been a priority for decades. Early investigations in mice showed that APAP is metabolized by cytochrome P450 enzymes to a reactive metabolite, now known as N-acetyl-p-benzoquinone imine (NAPQI), which can be detoxified by glutathione (GSH), and after hepatic GSH depletion, NAPQI binds to cellular proteins and initiates toxicity [5–7]. The identification of the critical role of GSH in protecting against APAP-induced cell death in preclinical models led to the development of the GSH precursor N-acetylcysteine (NAC) as clinical antidote against APAP poisoning [8]. More recently, the critical role of a mitochondrial oxidant stress and peroxynitrite formation was established in animals [9]. In addition, it was recognized that although the mitochondrial oxidant stress is initiated by protein adducts in mitochondria [10], activation of c-jun N-terminal kinase (JNK) in the cytosol and subsequent translocation of phospho-JNK to the mitochondria amplifies the formation of reactive oxygen and peroxynitrite in the mitochondrial matrix [11,12] leading to the opening of the mitochondrial membrane permeability transition pore (MPTP) and subsequent cell death [13]. This more detailed mechanistic insight resulted in the recognition of 4-methylpyrazole (Fomepizole), an effective inhibitor of cytochrome P450

2E1 (Cyp2E1) and JNK in mice, human hepatocytes and in patients [14–16], as a second possible clinical antidote against APAP poisoning [17,18].

Given that the mode of cell death is related to specific signaling events, substantial efforts were made to characterize the cell death after APAP overdose. Earlier studies in animals suggested necrosis or oncotic necrosis as the mode of cell death for drug- or chemical-induced liver injury due to cell and organelle swelling and release of cell contents [19]. It was thought that the cell death is caused by a catastrophic event, e.g., lipid peroxidation, calcium dysregulation, or protein adducts formation) [5–7,19,20]. However, over time more details emerged of the cellular signaling events resulting in APAP-induced necrosis [21–23], which led to the use of the term programmed necrosis [24,25]. During this time, it was also suggested that there might be substantial apoptotic cell death [26,27] but the lack of relevant caspase activation and cell shrinkage and absence of many other characteristics of apoptotic cell death did not support this hypothesis [28,29]. Necroptosis, a specific form of programmed necrosis, was also hypothesized to be involved in APAP-induced liver injury [30]. Although evidence for a role of receptor-interacting serine/threonine-protein 1 kinase (RIP1K) [30,31] and RIP3K [32,33] was provided, the fact that deficiency of mixed lineage kinase domain like pseudokinase (MLKL) did not protect against APAP hepatotoxicity [31], did not support necroptosis as a key cell death mode. In addition, ferroptosis, a cell death mechanism characterized by iron-dependent lipid peroxidation, was suggested to be involved in APAP-induced cell death [34]. However, under normal conditions, lipid peroxidation after APAP overdose is quantitatively insufficient to cause cell death and vitamin E does not protect [35]. Furthermore, lysosomal iron is taken up into mitochondria [36,37] and functions as a catalyst for peroxynitrite-mediated protein nitration critical for the mitochondrial MPTP opening, not lipid peroxidation [38]. This makes it unlikely that the key mode of cell death caused by APAP is ferroptosis [39]. More recently, it was suggested that the mode of cell death is actually pyroptosis [40–42], which raises the question whether there is any stronger evidence to support this newest idea for a mechanism of APAP hepatotoxicity than previously for apoptosis, necroptosis or ferroptosis.

## 2. Pyroptotic Cell Death Signaling

Pyroptosis is a necrotic programmed cell death mechanism that depends on gasdermin processing [43]. Although there are 6 known gasdermins, gasdermin D (GSDMD) is the prototypic gasdermin, which is present in most cell types including the liver [44]. GSDMD, as most other gasdermins, consists of a 31 kD N-terminal GSDMD$^{NT}$ fragment and a 22 kD C-terminal GSDMD$^{CT}$ fragment, which are connected through a linker region [44]. GSDMD$^{NT}$ forms pores in the cell membrane and GSDMD$^{CT}$ acts as a repressor of this pore formation [45–47]. The linker region contains caspase cleavage domains. Thus, based on the caspases involved, there is a canonical and a non-canonical inflammasome activation leading to pyroptosis [44] (Figure 1). In the canonical pathway, pathogen associated molecular patterns (PAMPs) or damage associated molecular patterns (DAMPs) activate cytosolic pattern recognition receptors including NOD-, LRR- and pyrin domain-containing protein 3 (NLRP3), which recruits the Apoptosis-associated speck-like protein containing a CARD (ASC) and pro-caspase-1. Caspase-1 is activated within the inflammasome and cleaves pro-IL-1β/pro-IL-18 to the active cytokine. In addition, caspase-1 also proteolytically releases GSDMD$^{NT}$ from GSDMD and the N-terminal fragment translocates to the plasma membrane and forms GSDMD$^{NT}$ pores [48]. These pores can trigger the release of IL-1β and IL-18 from the cell but eventually induce lytic cell death (pyroptosis) with extensive release of cellular contents through the ruptured cell membrane [44]. In contrast, the non-canonical inflammasome activation and pyroptosis is mediated by activation of caspase-11 (mice) or caspase 4 and 5 (humans) by lipopolysaccharide from Gram-negative bacteria. The active caspase-11 cleaves GSDMD, and the N-terminal fragments form a GSDMD$^{NT}$ pores. Cellular potassium release through the GSDMD$^{NT}$ pores triggers activation of the NLRP3 inflammasome with caspase 1 activation and pro-IL-1β/pro-IL-18 cleavage and

release. In addition, GSDMD<sup>NT</sup> pores induce the lytic cell death (pyroptosis) with cell swelling, plasma membrane rupture and extensive release of cellular contents [44].

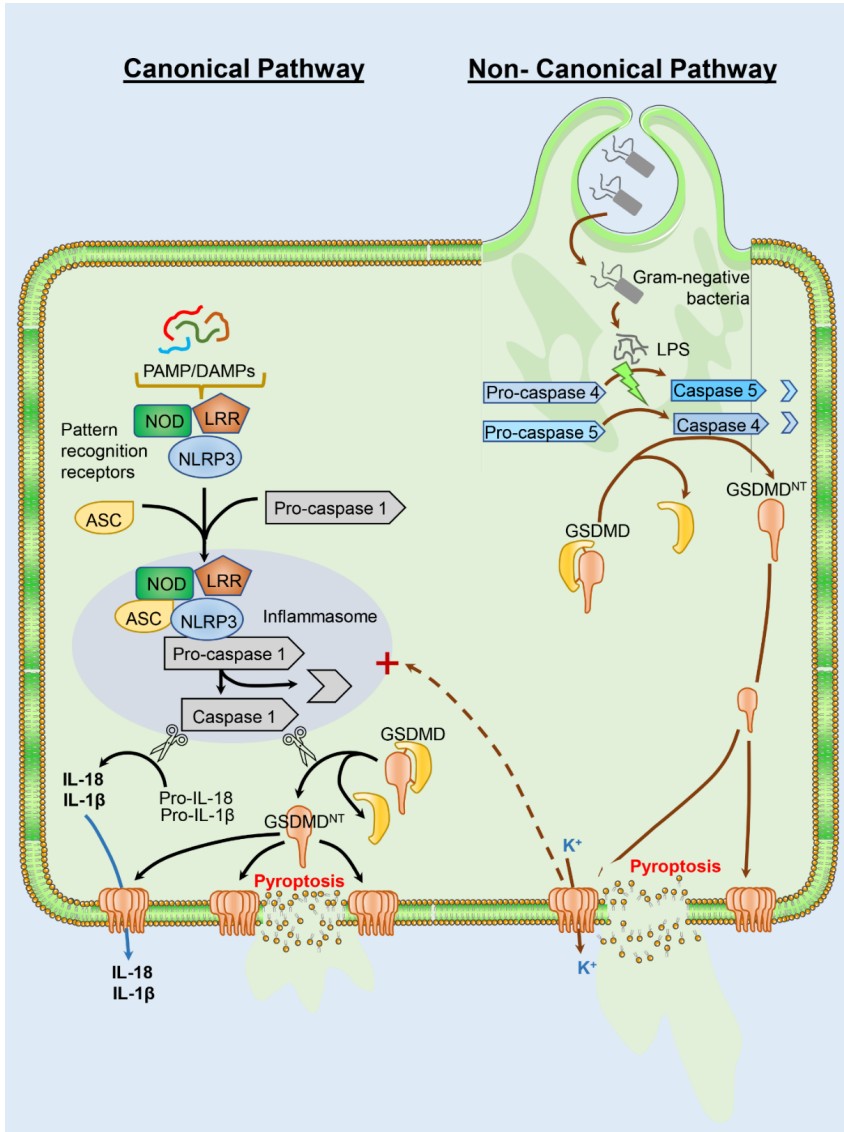

**Figure 1.** Pathways of pyroptosis. Gasdermin (GSDMD) cleavage and activation are central to pyroptotic cell death and can occur through a canonical or non-canonical pathway. The canonical pathway is initiated by recognition of pathogen associated molecular patterns (PAMPs) or damage associated molecular patterns (DAMPs) by cytosolic pattern recognition receptors such as NOD, LRR and NLRP3. These then recruit ASC and pro-caspase-1, which is activated to caspase-1 within the inflammasome. Caspase 1 cleaves pro-IL1β and pro-IL18 to the active cytokines, and also cleaves GSDMD to release the N-terminal domain (GSDMD<sup>NT</sup>), which translocates to the plasma membrane to form pores. While the active cytokines can be released through the GSDMD<sup>NT</sup> pores, these eventually cause lytic cell death with release of cellular contents through damaged cell membranes. In contrast to PAMPs and DAMPs, the non-canonical pyroptotic pathway is initiated by LPS from Gram-negative bacteria, which activates caspase 4 and 5 in humans. Activated caspase cleaves GSDMD as in the canonical pathway with formation of membrane pores, but cellular potassium release through the GSDMD<sup>NT</sup> pores can induce activation of the NLRP3 inflammasome for IL-1β and IL-18 release. Ultimately, the pores again induce lytic cell death due to release of cell contents.

In addition to the classical cleavage of GSDMD by caspase-1, or -11 (mice) and -4, -5 (humans), there is evidence for some crosstalk between apoptotic and pyroptotic signaling [44]. For example, caspase-8 activated during apoptosis, can also cleave GSDMD and promote pore formation, although at a slower rate [49]. However, active caspase-3 can cleave and inactive the N-terminal fragment [50,51]. Thus, GSDMD can trigger pyroptosis during apoptotic cell death, however, the physiological function of GSDMD activation/inactivation during apoptosis remains to be determined [44].

Despite the well-described mechanism of pyroptosis, GSDMD cleavage may not always end up with cell lysis. In some cell types, especially monocytes and neutrophils, the main function of gasdermin pores may be just the release of mature IL-1β and IL-18 in the absence of cell death [52,53]. These sublytic gasdermin pores can not only release the cytokines IL-1β and IL-18 but also other small cytosolic proteins [53,54]. Importantly, the membrane channels can release ions such as potassium, which triggers activation of the NLRP3 inflammasome after initiation of the non-canonical pathway by lipopolysaccharide [55,56]. An interesting question is what determines sublytic versus lytic gasdermin pore formation. Although cell type-specific expression of gasdermins and activation of caspases may be important factors, it was also recognized that GSDMD pores can be removed by an endosomal sorting complex required for transport (ESCRT) [57]. The ESCRT is recruited to the plasma membrane by calcium influx through the GSDMD pore and promotes the budding and release of vesicles with the GSDMD pore-containing membrane parts [58]. This process can completely repair the membrane damage or at least limit the function of GSDMD pores to the sublytic phase. Together, the process of pyroptosis is well regulated by cell type-specific expression of gasdermins, the degree of caspase activation and gasdermin cleavage and pore formation and the counteracting repair processes.

## 3. Evidence for Pyroptosis in APAP Toxicity

Recently, pyroptosis has been suggested to be a relevant cell death pathway in the standard murine model of APAP hepatotoxicity, i.e., 12–24 h after a dose of 300 mg/kg APAP in fasted mice [40–42]. Wang et al. provided evidence for cleavage of gasdermin D and of caspase-1 along with a minor increase in IL-1β and IL-18 in plasma. These changes were observed in both Kupffer cells and in hepatocytes. In addition, knock-down of peroxiredoxin 3 (Prdx3) aggravated the injury and all pyroptosis parameters, and knock-down of NLRP3 reduced all pyroptosis parameters and the overall liver injury [40]. The authors concluded that Prdx3 inhibits APAP-induced pyroptosis by attenuating the mitochondrial oxidant stress and preventing NLRP3 inflammasome activation [40]. However, there are many concerns with the interpretation of these experiments. First, the mitochondria-specific peroxidase Prdx3 protects against mitochondrial dysfunction by removing hydrogen peroxide and especially peroxynitrite [59,60]. It is well documented that peroxynitrite is the most important mitochondrial oxidant generated during APAP overdose [61]. APAP toxicity was greatly diminished by facilitating the synthesis of GSH as scavenger of peroxynitrite [62], the deficiency of the mitochondrial SOD2 dramatically enhanced peroxynitrite formation and injury [63,64] and the mitochondria-specific SOD mimetic mito-tempo eliminated peroxynitrite formation and effectively protected [65]. Thus, it is not surprising that Prdx3, an effective peroxynitrite scavenger located in mitochondria, reduced APAP toxicity [40]. However, the mitochondrial stress and dysfunction after an APAP overdose causes the MPT pore opening and nuclear DNA fragmentation leading to programmed necrosis not pyroptosis [23,66]. Second, the overall changes in pyroptotic parameters such as cleavage of gasdermin D or caspase 1 is generally below 2–3-fold of baseline, which leads to very limited IL-1β and IL-18 formation [40]. In fact, the increase of plasma IL-1β levels from <10 pg/mL in controls to 20 pg/mL after APAP reported by the authors [40] is almost identical to levels published previously in a similar mouse model [67]. However, the minor changes in IL-1β were insufficient to impact the injury [67]. Moreover, a recent study did not observe protection against APAP or thioacetamide hepatotoxicity in gasdermin D- and E-knock-out mice [68]. Interestingly, this study also demonstrated that thioacetamide toxic-

ity did not involve gasdermin D or E cleavage, but no data were provided on gasdermin cleavage after APAP [68]. In contrast, another report indicated increased APAP-induced liver injury in gasdermin D-deficient mice [69]. However, as discussed [25], this alleged protection by gasdermin D may have been the result of a substrain mismatch between the KO mice and the wildtype animals.

More recently, 2 additional manuscripts were published with the conclusion that pyroptosis may be a mode of cell death during APAP hepatotoxicity in mice [41,42]. The conclusion was based on only minor (<2-fold) increases in gasdermin-D, caspase-1 and NLRP3 protein expression at the whole liver level [41,42]. However, minor changes in protein expression are insufficient to explain extensive cellular necrosis and are certainly not specific for pyroptotic cell death. Furthermore, no specific intervention for pyroptosis was used to justify the conclusion that pyroptosis may be relevant for APAP-induced liver injury. Thus, most of the data and conclusions regarding the critical role of pyroptosis in APAP-induced liver injury are based on correlations and have to be questioned.

## 4. Evidence against Pyroptosis in Acetaminophen-Induced Liver Injury

As discussed above, key signaling events in pyroptotic cell death are the activation of the NLRP3 inflammasome with cleavage of pro-caspase 1. The active caspase cleaves gasdermin and the N-terminal fragments translocate to the cell membrane forming a pore to facilitate the release of IL-1β, and eventually cause cell death. A recent study showed that gasdermin D- and E-deficient mice are not protected against APAP hepatotoxicity [68]. Caspase 1 is one of many caspases that are all effectively inhibited by suicide substrates like Z-VAD-fmk. These caspase inhibitors bind irreversibly to the active caspase and block the enzyme activity [70]). Pancaspase inhibitors and inhibitors more specific for individual caspases, e.g., caspase 3 or -8, are highly effective in models of hepatocellular apoptosis such as galactosamine/endotoxin shock [71–73], or agonistic Fas antibody-induced liver injury [74–76] but fail to protect against APAP hepatotoxicity [28,67,77,78]. The few studies that claim a beneficial effect of pancaspase inhibitors in the APAP model [27,79] were later shown to be effects of the solvent dimethyl sulfoxide (DMSO) [78,80], which is an effective inhibitor of cytochrome P450 enzymes [81]. Nevertheless, when IL-1β formation was evaluated after an APAP overdose, there was a significant but moderate increase of IL-1β mRNA and mature IL-1β protein formation [67,82] suggesting activation of the inflammasome and caspase-1 in the mouse model. Indeed, a pancaspase inhibitor reduced IL-1β protein levels but did not affect IL-1β mRNA levels [67]. However, the IL-1β levels were far below concentrations that could have affected APAP-induced liver injury both in mice [67,83] and in human overdose patients [84], which is consistent with the fact that IL-1 receptor-deficient mice are not protected against APAP hepatotoxicity [67].

The sterile inflammatory response after an APAP overdose is initiated by the release of DAMPs from necrotic hepatocytes including mitochondrial DNA and nuclear DNA fragments [85], which bind to Toll-like receptors, e.g., TLR9, on Kupffer cells and trigger the transcriptional activation of pro-inflammatory cytokine genes [82,86]. Additional DAMPs like ATP activate the NLRP3 inflammasome by binding to the purine receptor P2X7 on Kupffer cells, which causes the activation of caspase-1 and processing of pro-IL-1β to the active cytokine [48]. In addition to the relatively moderate formation of these active cytokines, the fundamental issue is that the activation of the inflammasome and caspase-1 and IL-1β formation occurs mainly in Kupffer cells [83] whereas the main APAP-induced cell death occurs in hepatocytes. This would question the possibility that caspase-mediated pyroptosis could be responsible for APAP-induced hepatocyte necrosis. Consistent with these findings, analysis of time series single-cell RNA sequencing data during APAP hepatotoxicity reveals that Kupffer cells and infiltrating macrophages are the primary cells expressing the pyroptotic genes Casp1, IL-1β, Gsdmd and Nlrp3 while hepatocytes, regardless of spatial location within the liver lobule, maintain very low expression levels for these pyroptotic genes (Figure 2) [87–89]. A limitation of the single cell data is that it cannot distinguish between the original Kupffer cells and monocyte-derived macrophages

that may have shifted their phenotype during the resolution phase of the injury. However, this does not affect our conclusion that genes related to pyroptosis are primarily located in macrophages and not in hepatocytes, which means that hepatocytes are unlikely to be able to die by pyroptosis. Collectively, these findings support a potential role of pyroptosis to resolve the sterile inflammatory response after an APAP overdose, however, pyroptosis as the primary mode of hepatocyte cell death is unlikely.

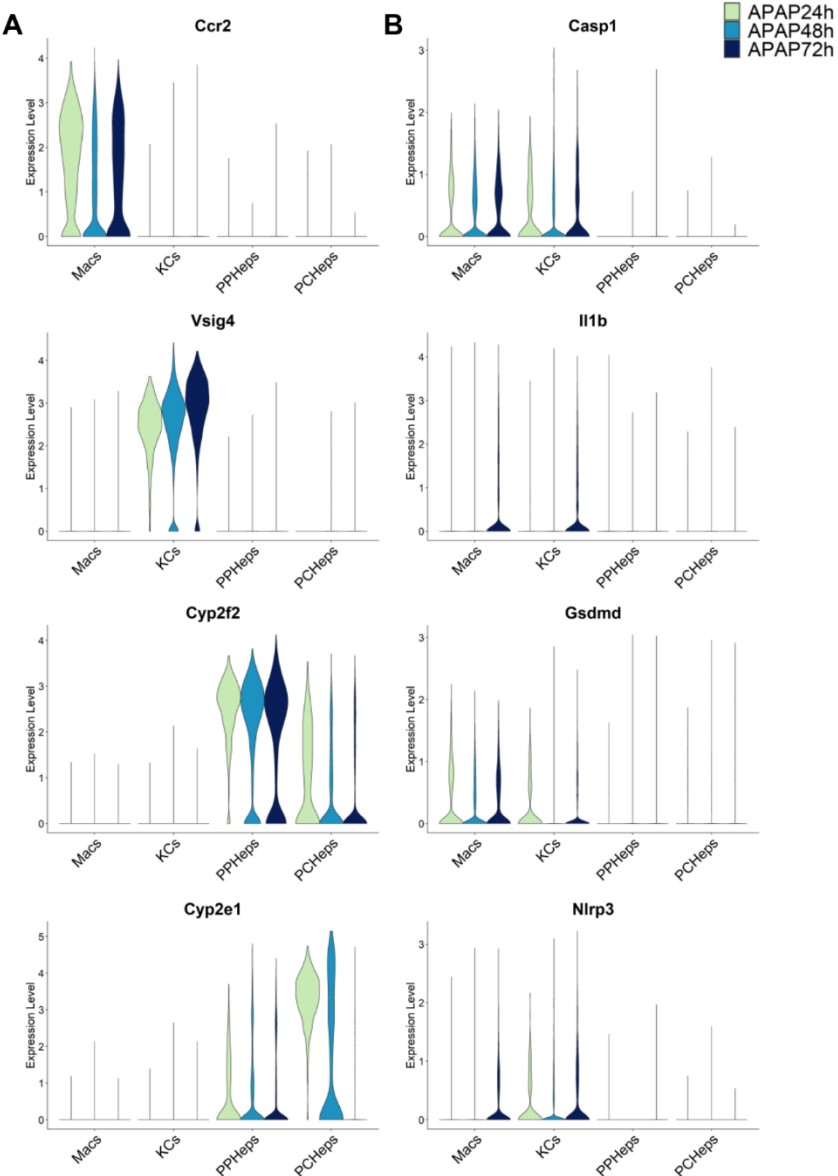

**Figure 2.** Single-cell RNA sequencing data reveal pyroptotic genes are expressed predominantly in infiltrating macrophages (Macs) or Kupffer cells (KCs) during the acetaminophen time course (24, 48, 72 h) in mice. Data in the left column (**A**) show representative cell type markers for infiltrating macrophages (Ccr2), KCs (Vsig4), periportal hepatocytes (cyp2f2) and pericentral hepatocytes (cyp2e1). Data in the right column (**B**) show canonical genes of pyroptosis (Casp1, IL-1β, Gsdmd, and Nlrp3). Single-cell RNA seq data were aggregated from publicly available datasets (GSE136679, GSE200771, zenodo.6035873).

## 5. Summary and Conclusions

The mode of cell death after an APAP overdose has been extensively investigated and discussed [24,25]. Besides oncotic necrosis, other forms of cell death including apoptosis, necroptosis, ferroptosis and most recently pyroptosis have been implicated in APAP-

induced liver injury. However, one glaring weakness of many studies claiming one of these cell death pathways as central to the toxicity is the lack of discussion of other cell death pathways and the fact that others have come to fundamentally different conclusions. A problem in this respect is that there are overlap in signaling pathways between different modes of cell death (discussed in detail: [25,90]), which can lead to misinterpretations. For example, the translocation of Bax to the mitochondria and formation of pores in the outer mitochondrial membrane, which can induce release of cytochrome c and other inter-membrane proteins, is generally considered a pro-apoptotic signaling event [91]. However, mitochondrial Bax translocation is an early event in APAP hepatotoxicity that facilitates the release of endonuclease G and apoptosis-inducing factor (AIF) from mitochondria and promotes nuclear DNA fragmentation [92]. Thus, Bax pores amplify caspase activation through mitochondrial cytochrome c release during apoptosis but promote nuclear DNA degradation through mitochondrial endonuclease G release and its translocation to the nucleus during APAP-induced necrosis. Importantly, these events do not require transcriptional activation of Bax. It is therefore important to not just associate the increased expression of mRNAs or proteins of certain genes presumably associated with a cell death pathway but fully understand the functional importance of the genes in the signaling events of cell death. For pyroptosis, this means that neither increased protein expression of inflammasome components, caspase-1 nor gasdermins are evidence for pyroptotic cell death. In contrast, activation of caspase-1, cleavage of gasdermins and translocation of the N-terminal fragment to the cell membrane and pore formation are the critical events [44]. Although evidence was provided for modest caspase-1 and gasdermin D cleavage and limited formation of IL-1β and IL-18 in both hepatocytes and Kupffer cells after APAP overdose [40], gasdermin D gene knockout mice were not protected against APAP toxicity [68,69] suggesting no relevant pyroptosis in APAP-induced liver injury. However, based on the experience with apoptosis [29,93] and ferroptosis [39] in APAP hepatotoxicity, questionable evidence (minor increases of mRNAs or protein expression of genes presumably related to pyroptosis) can be expected to be used by other authors in the future as evidence for pyroptosis. Unfortunately, this will only add to the confusion but not advance our understanding of the pathophysiology of APAP or the mechanism of protection with various interventions. We hope that our discussion can limit the casual use of cell death modes like pyroptosis based on gene expression data and encourage proper investigations into this cell death pathway using specific interventions, e.g., gasdermin D gene knockout mice, and documentation of quantitively relevant inflammasome activation and gasdermin pore formation in specific cell types of the liver. Furthermore, authors should integrate their new findings with earlier available information on necrotic cell signaling after an APAP overdose in order to make progress in understanding the nuances of APAP pathophysiology.

**Author Contributions:** H.J. conceived the manuscript and wrote the first draft. A.R. edited the manuscript and generated the graphic. D.S.U. edited the manuscript and performed the bioinformatic analysis of scRNA seq data and generated the figure. All authors have read and agreed to the published version of the manuscript.

**Funding:** The work discussed was supported in part by the U.S. National Institutes of Health grants R01 DK102142 (H.J.) and R01 DK125465 (A.R).

**Data Availability Statement:** Single-cell RNA seq data were aggregated from publicly available datasets (GSE136679, GSE200771, zenodo.6035873).

**Conflicts of Interest:** The authors declare no conflict of interest.

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
