# Peer review of "Role of Pyroptosis in Acetaminophen-Induced Hepatotoxicity"

_livers, doi:10.3390/livers2040032_

Round 1

Reviewer 1 Report

This review nicely summarizes prior work on the role of pyroptosis in acetaminophen induced liver injury.  Overall, and as stated by the authors, it is clear from prior studies utilizing mice with genetic deletions in the pyroptotic pathway that pyroptosis is not critical for acetaminophen-induced hepatocyte injury.     

1)      In the single cell RNA sequencing analysis, it is possible that at the later times (48 and 72 hours), many of the VSIG4-expressing, Ccr2-negative cells are infiltrating monocytes that have shifted phenotype during the resolution phase of injury.  This does not take away from the conclusion, however, that pyroptosis-associated mRNAs are expressed at greater levels in macrophages when compared to hepatocytes.

Author Response

We thank the reviewer for his/her constructive comments and modified the manuscript accordingly.

Reviewer 1:

This review nicely summarizes prior work on the role of pyroptosis in acetaminophen induced liver injury.  Overall, and as stated by the authors, it is clear from prior studies utilizing mice with genetic deletions in the pyroptotic pathway that pyroptosis is not critical for acetaminophen-induced hepatocyte injury.    

Comment 1: In the single cell RNA sequencing analysis, it is possible that at the later times (48 and 72 hours), many of the VSIG4-expressing, Ccr2-negative cells are infiltrating monocytes that have shifted phenotype during the resolution phase of injury.  This does not take away from the conclusion, however, that pyroptosis-associated mRNAs are expressed at greater levels in macrophages when compared to hepatocytes.

Response:  We agree, the single cell data cannot distinguish between the original Kupffer cells and monocyte-derived macrophages that may have shifted their phenotype during the resolution phase of the injury. However, as the reviewer pointed out, this does not affect our conclusion that genes related to pyroptosis are primarily located in macrophages and not in hepatocytes, which means that hepatocytes are unlikely to be able to die by pyroptosis. We added clarifying sentences on p. 13.

Reviewer 2 Report

The paper by Hartmut Jaeschke and colleagues proposes to review the role of pyroptosis in acetaminophen-induced hepatotoxicitye. Because of the wide-spread availability of APAP, intentional and accidental overdosing is relatively common and is responsible for most acute liver failure. Understanding the mechanisms of APAP-induced liver injury has important clinical significance. It makes the subject of the review highly pertinent.

Please check out every single gene/protein name and use the full name when it was first mentioned. For example: RIP1K, MLKL, MPTP...

  1. The author should better define the difference and crosstalk between pyroptosis and other forms of cell death (oncotic necrosis, apoptosis, necroptosis, ferroptosis).
  2. In Part2, Pyroptotic cell death signaling. Due to the bidirectional crosstalk among different forms of cell death. The author should highlight common and unique pathways and their effect.

The authors should better define throughout the text when they are referring data observed in humans, animal models or in vitro studies.

Author Response

We thank the reviewer for his/her constructive comments and modified the manuscript accordingly.

The paper by Hartmut Jaeschke and colleagues proposes to review the role of pyroptosis in acetaminophen-induced hepatotoxicity. Because of the wide-spread availability of APAP, intentional and accidental overdosing is relatively common and is responsible for most acute liver failure. Understanding the mechanisms of APAP-induced liver injury has important clinical significance. It makes the subject of the review highly pertinent.

Comment 1:  Please check out every single gene/protein name and use the full name when it was first mentioned. For example: RIP1K, MLKL, MPTP...

Response: As suggested, we have now provided the full names of all genes and proteins when first mentioned in the text.

Comment 2:  The author should better define the difference and crosstalk between pyroptosis and other forms of cell death (oncotic necrosis, apoptosis, necroptosis, ferroptosis).

Response: We agree that differences and potential crosstalks between different modes of cell death are interesting and important. However, this discussion is somewhat outside the scope of this review, which is to specifically assess the relevance of pyroptosis in acetaminophen hepatotoxicity. We have highlighted key features of other forms of cell death in the Introduction and have now added a potentially relevant example of apoptosis/pyroptosis crosstalk in the discussion of the pyroptosis pathways (p. 6).

Comment 3: In Part2, Pyroptotic cell death signaling. Due to the bidirectional crosstalk among different forms of cell death. The author should highlight common and unique pathways and their effect.

Response: As mentioned above, the objective of this review was not to discuss in depth signaling pathways of all cell death modes and their potential crosstalk, but to focus on the potential role of pyroptosis in acetaminophen-induced cell death. However, we have now added an example of crosstalk between apoptosis and pyroptosis (p. 6) and we discussed overlap of signaling pathways between apoptosis and necrosis (p. 15). For more in-depth coverage of interactions between cell death modes the reader is now referred to recent reviews on that topic (p. 15/16).

Comment 4: The authors should better define throughout the text when they are referring data observed in humans, animal models or in vitro studies.

Response: As suggested by the reviewer, we have clarified throughout the manuscript whether the statements refer to studies in mice, human hepatocytes or patients.